# Recursive Training of 2D-3D Convolutional Networks for Neuronal Boundary Detection

**Kisuk Lee, Aleksandar Zlateski**
Massachusetts Institute of Technology
{kisuklee,zlateski}@mit.edu

**Ashwin Vishwanathan, H. Sebastian Seung**
Princeton University
{ashwinv,sseung}@princeton.edu

## Abstract

Efforts to automate the reconstruction of neural circuits from 3D electron microscopic (EM) brain images are critical for the field of connectomics. An important computation for reconstruction is the detection of neuronal boundaries. Images acquired by serial section EM, a leading 3D EM technique, are highly anisotropic, with inferior quality along the third dimension. For such images, the 2D maxpooling convolutional network has set the standard for performance at boundary detection. Here we achieve a substantial gain in accuracy through three innovations. Following the trend towards deeper networks for object recognition, we use a much deeper network than previously employed for boundary detection. Second, we incorporate 3D as well as 2D filters, to enable computations that use 3D context. Finally, we adopt a recursively trained architecture in which a first network generates a preliminary boundary map that is provided as input along with the original image to a second network that generates a final boundary map. Backpropagation training is accelerated by ZNN, a new implementation of 3D convolutional networks that uses multicore CPU parallelism for speed. Our hybrid 2D-3D architecture could be more generally applicable to other types of anisotropic 3D images, including video, and our recursive framework for any image labeling problem.

## 1   Introduction

Neural circuits can be reconstructed by analyzing 3D brain images from electron microscopy (EM). Image analysis has been accelerated by semiautomated systems that use computer vision to reduce the amount of human labor required [1, 2, 3]. However, analysis of large image datasets is still laborious [4], so it is critical to increase automation by improving the accuracy of computer vision algorithms.

A variety of machine learning approaches have been explored for the 3D reconstruction of neurons, a problem that can be formulated as image segmentation or boundary detection [5, 6]. This paper focuses on neuronal boundary detection in images from serial section EM, the most widespread kind of 3D EM [7]. The technique starts by cutting and collecting ultrathin (30 to 100 nm) sections of brain tissue. A 2D image is acquired from each section, and then the 2D images are aligned. The spatial resolution of the resulting 3D image stack along the $z$ direction (perpendicular to the cutting plane) is set by the thickness of the sections. This is generally much worse than the resolution that EM yields in the $xy$ plane. In addition, alignment errors may corrupt the image along the $z$ direction.

Due to these issues with the $z$ direction of the image stack [6, 8], most existing analysis pipelines begin with 2D processing and only later transition to 3D. The stages are: (1) neuronal boundary detection within each 2D image, (2) segmentation of neuron cross sections within each 2D image, and (3) 3D reconstruction of individual neurons by linking across multiple 2D images [1, 9].

Boundary detection in serial section EM images is done by a variety of algorithms. Many algorithms were compared in the ISBI'12 2D EM segmentation challenge, a publicly available dataset and benchmark [10]. The winning submission was an ensemble of max-pooling convolutional networks (ConvNets) created by IDSIA [11]. One of the ConvNet architectures shown in Figure 1 (N4) is the largest architecture from [11], and serves as a performance baseline for the research reported here.

We improve upon N4 by adding several new elements (Fig. 1):

**Increased depth**     Our VD2D architecture is deeper than N4 (Figure 1), and borrows other now-standard practices from the literature, such as rectified linear units (ReLUs), small filter sizes, and multiple convolution layers between pooling layers. VD2D already outperforms N4, without any use of 3D context. VD2D is motivated by the principle "the deeper, the better," which has become popular for ConvNets applied to object recognition [12, 13].

**3D as well as 2D**     When human experts detect boundaries in EM images, they use 3D context to disambiguate certain locations. VD2D3D is also able to use 3D context, because it contains 3D filters in its later layers. ConvNets with 3D filters were previously applied to block face EM images [2, 3, 14]. Block face EM is another class of 3D EM techniques, and produces nearly isotropic images, unlike serial section EM. VD2D3D also contains 2D filters in its earlier layers. This novel hybrid use of 2D and 3D filters is suited for the highly anisotropic nature of serial section EM images.

**Recursive training of ConvNets**     VD2D and VD2D3D are concatenated to create an extremely deep network. The output of VD2D is a preliminary boundary map, which is provided as input to VD2D3D in addition to the original image (Fig. 1). Based on these two inputs, VD2D3D is trained to compute the final boundary map. Such "recursive" training has previously been applied to neural networks for boundary detection [8, 15, 16], but not to ConvNets.

**ZNN for 3D deep learning** Very deep ConvNets with 3D filters are computationally expensive, so an efficient software implementation is critical. We trained our networks with ZNN (`https://github.com/seung-lab/znn-release`, [17]), which uses multicore CPU parallelism for speed. ZNN is one of the few deep learning implementations that is well-optimized for 3D.

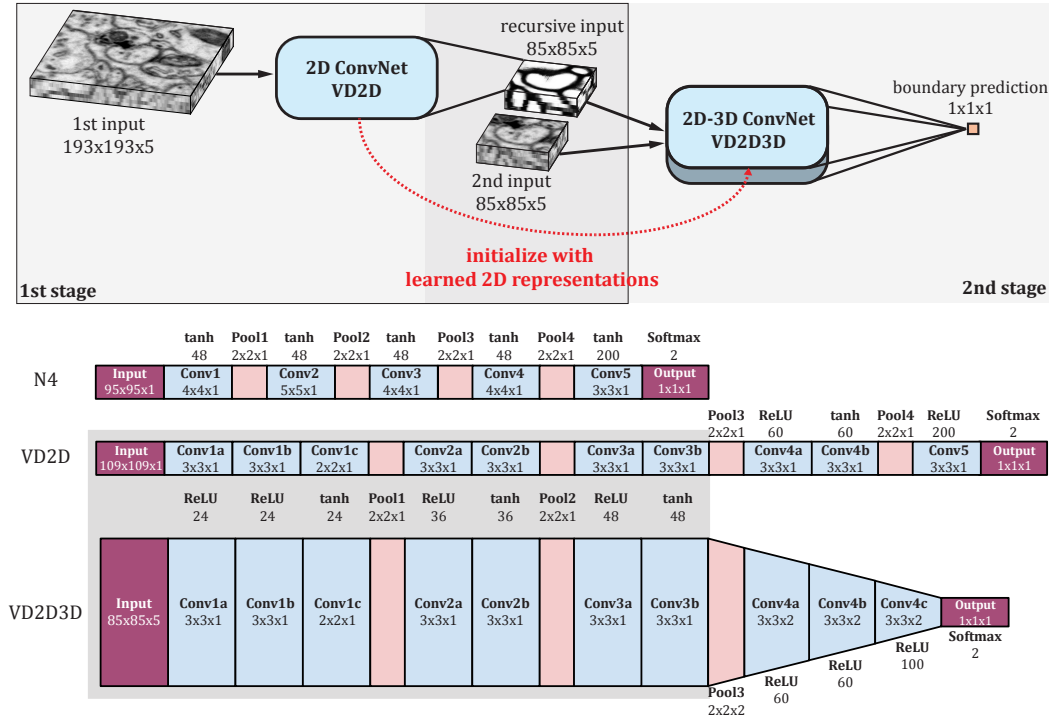

Figure 1: An overview of our proposed framework (top) and model architectures (bottom). The number of trainable parameters in each model is 220K (N4), 230K (VD2D), 310K (VD2D3D).

While we have applied the above elements to serial section EM images, they are likely to be generally useful for other types of images. The hybrid use of 2D and 3D filters may be useful for video, which can also be viewed as an anisotropic 3D image. Previous 3D ConvNets applied to video processing [18, 19] have used 3D filters exclusively.

Recursively trained ConvNets are potentially useful for any image labeling problem. The approach is very similar to recurrent ConvNets [20], which iterate the same ConvNet. The recursive approach uses different ConvNets for the successive iterations. The recursive approach has been justified in several ways. In MRF/CRF image labeling, it is viewed as the sequential refinement of the posterior probability of a pixel being assigned a label, given both an input image and recursive input from the previous step [21]. Another viewpoint on recursive training is that statistical dependencies in label (category) space can be directly modeled from the recursive input [15]. From the neurobiological viewpoint, using a preliminary boundary map for an image to guide the computation of a better boundary map for the image can be interpreted as employing a top-down or attentional mechanism.

We expect ZNN to have applications far beyond the one considered in this paper. ZNN can train very large networks, because CPUs can access more memory than GPUs. Task parallelism, rather than the SIMD parallelism of GPUs, allows for efficient training of ConvNets with arbitrary topology. A self-tuning capability automatically optimizes each layer by choosing between direct and FFT-based convolution. FFT convolution may be more efficient for wider layers or larger filter size [22, 23]. Finally, ZNN may incur less software development cost, owing to the relative ease of the general-purpose CPU programming model.

Finally, we applied our ConvNets to images from a new serial section EM dataset from the mouse piriform cortex. This dataset is important to us, because we are interested in conducting neuroscience research concerning this brain region. Even to those with no interest in piriform cortex, the dataset could be useful for research on image segmentation algorithms. Therefore we make the annotated dataset publicly available (`http://seunglab.org/data/`).

## 2    Dataset and evaluation

**Images of mouse piriform cortex**    The datasets described here were acquired from the piriform cortex of an adult mouse prepared with aldehyde fixation and reduced osmium staining [24]. The tissue was sectioned using the automatic tape collecting ultramicrotome (ATUM) [25] and sections were imaged on a Zeiss field emission scanning electron microscope [26]. The 2D images were assembled into 3D stacks using custom MATLAB routines and TrakEM2, and each stack was manually annotated using VAST (`https://software.rc.fas.harvard.edu/lichtman/vast/`, [25]) (Figure 2). Then each stack was checked and corrected by another annotator.

The properties of the four image stacks are detailed in Table 1. It should be noted that image quality varies across the stacks, due to aging of the field emission source in the microscope. In all experiments we used `stack1` for testing, `stack2` and `stack3` for training, and `stack4` as an additional training data for recursive training.

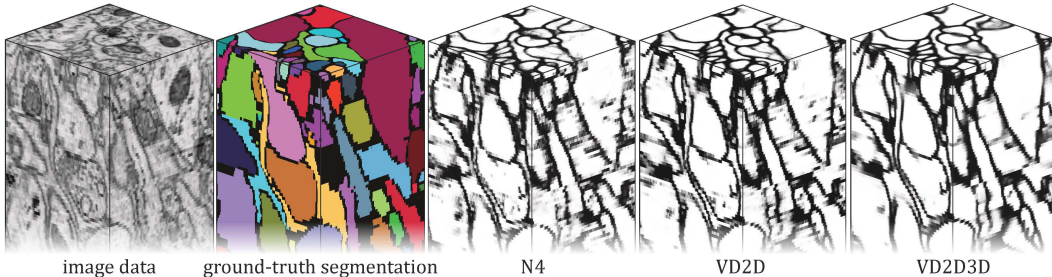

|  image data | ground-truth segmentation | N4 | VD2D | VD2D3D |

Figure 2: Example dataset (`stack1`, Table 1) and results of each architecture on `stack1`.

Table 1: Piriform cortex datasets

| Name | stack1 | stack2 | stack3 | stack4 |
|---|---|---|---|---|
| Resolution (nm$^3$) | $7 \cdot 7 \cdot 40$ | $7 \cdot 7 \cdot 40$ | $7 \cdot 7 \cdot 40$ | $10 \cdot 10 \cdot 40$ |
| Dimension (voxel$^3$) | $255 \cdot 255 \cdot 168$ | $512 \cdot 512 \cdot 170$ | $512 \cdot 512 \cdot 169$ | $256 \cdot 256 \cdot 121$ |
| # samples | 10.9M | 44.6 M | 44.3 M | 7.9 M |
| Usage | Test | Training | Training | Training (extra) |

**Pixel error**    We use softmax activation in the output layer of our networks to produce per-pixel real-valued outputs between 0 and 1, each of which is interpreted as the probability of an output pixel being boundary, or vice versa. This real-valued "boundary map" can be thresholded to generate a binary boundary map, from which the pixel-wise classification error is computed. We report the best classification error obtained by optimizing the binarization threshold with line search.

**Rand score**    We evaluate 2D segmentation performance with the Rand scoring system [27, 28]. Let $n_{ij}$ denote the number of pixels simultaneously in the $i^{\text{th}}$ segment of the proposal segmentation and the $j^{\text{th}}$ segment of the ground truth segmentation. The Rand merge score and the Rand split score

$$V_{\text{merge}}^{\text{Rand}} = \frac{\sum_{ij} n_{ij}^2}{\sum_i (\sum_j n_{ij})^2}, \quad V_{\text{split}}^{\text{Rand}} = \frac{\sum_{ij} n_{ij}^2}{\sum_j (\sum_i n_{ij})^2}.$$

are closer to one when there are fewer merge and split errors, respectively. The Rand F-score is the harmonic mean of $V_{\text{merge}}^{\text{Rand}}$ and $V_{\text{split}}^{\text{Rand}}$.

To compute the Rand scores, we need to first obtain 2D neuronal segmentation based on the real-valued boundary map. To this end, we apply two segmentation algorithms with different levels of sophistication: (1) simple thresholding followed by computing 2D connected components, and (2) modified graph-based watershed algorithm [29]. We report the best Rand F-score obtained by optimizing parameters for each algorithm with line search, as well as the precision-recall curve for the Rand scores.

## 3   Training with ZNN

ZNN [17] was built for 3D ConvNets. 2D convolution is regarded as a special case of 3D convolution, in which one of the three filter dimensions has size 1. For the details on how ZNN implements task parallelism on multicore CPUs, we refer interested readers to [17]. Here we describe only aspects of ZNN that are helpful for understanding how it was used to implement the ConvNets of this paper.

**Dense output with maximum filtering**    In object recognition, a ConvNet is commonly applied to produce a single output value for an entire input image. However, there are many applications in which "dense output" is required, i.e., the ConvNet should produce an output image with the same resolution as the original input image. Such applications include boundary detection [11], image labeling [30], and object localization [31].

ZNN was built from the ground up for dense output and also for dense feature maps.[1] ZNN employs max-filtering, which slides a window across the image and applies the maximum operation to the window (Figure 3). Max-filtering is the dense variant of max-pooling. Consequently all feature maps remain intact as dense 3D volumes during both forward and backward passes, making them straightforward for visualization and manipulation.

On the other hand, all filtering operations are sparse, in the sense that the sliding window samples sparsely from a regularly spaced set of voxels in the image (Figure 3). ZNN can control the spacing/sparsity of any filtering operation, either convolution or max-filtering.

ZNN can efficiently compute the dense output of a sliding window max-pooling ConvNet by making filter sparsity depend on the number of prior max-filterings. More specifically, each max-filtering

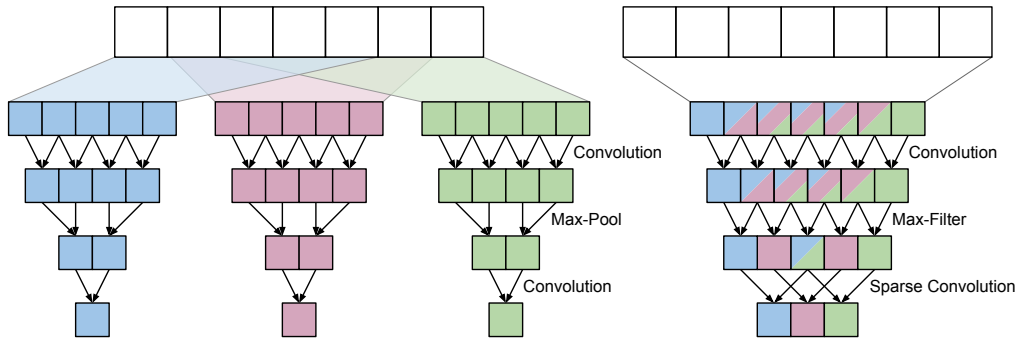

Figure 3: Sliding window max-pooling ConvNet (left) applied on three color-coded adjacent input windows producing three outputs. Equivalent outputs produced by a max-filtering ConvNet with sparse filters (right) applied on a larger window. Computation is minimized by reusing the intermediate values for computing multiple outputs (as color coded).

increases the sparsity of all subsequent filterings by a factor equal to the size of the max-pooling window. This approach, which we employ for the paper, is also called "skip-kernels" [31] or "filter rarefaction" [30], and is equivalent in its results to "max-fragmentation-pooling" [32, 33]. Note however that ZNN is more general, as the sparsity of filters need not depend on max-filtering, but can be controlled independently.

**Output patch training**    Training in ZNN is based on loss computed over a dense output patch of arbitrary size. The patch can be arbitrarily large, limited only by memory. This includes the case of a patch that spans the entire image [30, 33]. Although large patch sizes reduce the computational cost per output pixel, neighboring pixels in the patch may provide redundant information. In practice, we choose an intermediate output patch size.

## 4    Network architecture

**N4**    As a baseline for performance comparisons, we adopted the largest 2D ConvNet architecture (named N4) from Cireşan et al. [11] (Figure 1).

**VD2D**    The architecture of VD2D ("Very Deep 2D") is shown in Figure 1. Multiple convolution layers are between each max-pooling layer. All convolution filters are $3 \times 3 \times 1$, except that `Conv1c` uses a $2 \times 2 \times 1$ filter to make the "field of view" or "receptive field" for a single output pixel have an odd-numbered size and therefore centerable around the output pixel. Due to the use of smaller filters, the number of trainable parameters in VD2D (230K) is roughly the same as in the shallower N4 (220K).

**VD2D3D**    The architecture of VD2D3D ("Very Deep 2D-3D") is initially identical to VD2D (Figure 1), except that later convolution layers switch to $3 \times 3 \times 2$ filters. This causes the number of trainable parameters to increase, so we compensate by trimming the size of `Conv4c` to just 100 feature maps. The 3D filters in the later layers should enable the network to use 3D context to detect neuronal boundaries. The use of 2D filters in the initial layers makes the network faster to run and train.

**Recursive training**    It is possible to apply VD2D3D by itself to boundary detection, giving the raw image as the only input. However, we use a recursive approach in which VD2D3D receives an extra input, the output of VD2D. As we will see below, this produces a significant improvement in performance. It should be noted that instead of providing the recursive input directly to VD2D3D, we added new layers [2] dedicated to processing it. This separate, parallel processing stream for recursive input joins the main stream at `Conv1c`, allowing for more complex, highly nonlinear interaction between the low-level features and the contextual information in the recursive input.

# 5 Training procedures

Networks were trained using backpropagation with the cross-entropy loss function. We first trained VD2D, and then trained VD2D3D. The 2D layers of VD2D3D were initialized using trained weights from VD2D. This initialization meant that our recursive approach bore some similarity to recurrent ConvNets, in which the first and second stage networks are constrained to be identical [20]. However, we did not enforce exact weight sharing, but fine-tuned the weights of VD2D3D.

**Output patch**    As mentioned earlier, training with ZNN is done by dense output patch-based gradient update with per-pixel loss. During training, an output patch of specified size is randomly drawn from the training stacks at the beginning of each forward pass.

**Class rebalancing**    In dense output patch-based training, imbalance between the number of training samples in different classes (e.g. boundary/non-boundary) can be handled by either sampling a balanced number of pixels from an output patch, or by differentially weighting the per-pixel loss [30]. In our experiment, we adopted the latter approach (loss weighting) to deal with the high imbalance between boundary and non-boundary pixels.

**Data augmentation**    We used the same data augmentation method used in [11], randomly rotating and flipping 2D image patches.

**Hyperparameter**    We always used the fixed learning rate of 0.01 with the momentum of 0.9. When updating weights we divided the gradient by the total number of pixels in an output patch, similar to the typical minibatch averaging.

We first trained N4 with an output patch of size $200 \times 200 \times 1$ for 90K gradient updates. Next, we trained VD2D with $150 \times 150 \times 1$ output patches, reflecting the increased size of model compared to N4. After 60K updates, we evaluated the trained VD2D on the training stacks to obtain preliminary boundary maps, and started training VD2D3D with $100 \times 100 \times 1$ output patches, again reflecting the increased model complexity. We trained VD2D3D for 90K updates. In this recursive training stage we additionally used `stack4` to prevent VD2D3D from being overly dependent on the good-quality boundary maps for training stacks. It should be noted that `stack4` has slightly lower $xy$-resolution than other stacks (Table 1), which we think is helpful in terms of learning multi-scale representation.

Our proposed recursive framework is different from the training of recurrent ConvNets [20] in that recursive input is not dynamically produced by the first ConvNet during training, but evaluated before and being fixed throughout the recursive training stage. However, it is also possible to further train the first ConvNet even after evaluating its preliminary output as recursive input to the second ConvNet. We further trained VD2D for another 30K updates while VD2D3D is being trained. We report the final performance of VD2D after a total of 90K updates. We also replaced the initial VD2D boundary map with the final one when evaluating VD2D3D results. With ZNN, it took two days to train both N4 and VD2D for 90K updates, and three days to train VD2D3D for 90K updates.

# 6 Results

In this section, we show both quantitative and qualitative results obtained by the three architectures shown in Figure 1, namely N4, VD2D, and VD2D3D. The pixel-wise classification error of each model on test set was $10.63\%$ (N4), $9.77\%$ (VD2D), and $8.76\%$ (VD2D3D).

**Quantitative comparison**    Figure 4 compares the result of each architecture on test set (`stack1`), both quantitatively and qualitatively. The leftmost bar graph shows the best 2D Rand F-score of each model obtained by 2D segmentation with (1) simpler connected component clustering and (2) more sophisticated watershed-based segmentation. The middle and rightmost graphs show the precision-recall curve of each model for the Rand scores obtained with the connected component and watershed-based segmentation, respectively. We observe that VD2D performs significantly better than N4, and also VD2D3D outperforms VD2D by a significant margin in terms of both best Rand F-score and overall precision-recall curve.

**Qualitative comparison**    Figure 2 shows the visualization of boundary detection results of each model on test set, along with the original EM images and ground truth boundary map. We observe that false detection of boundary on intracellular regions was significantly reduced in VD2D3D,

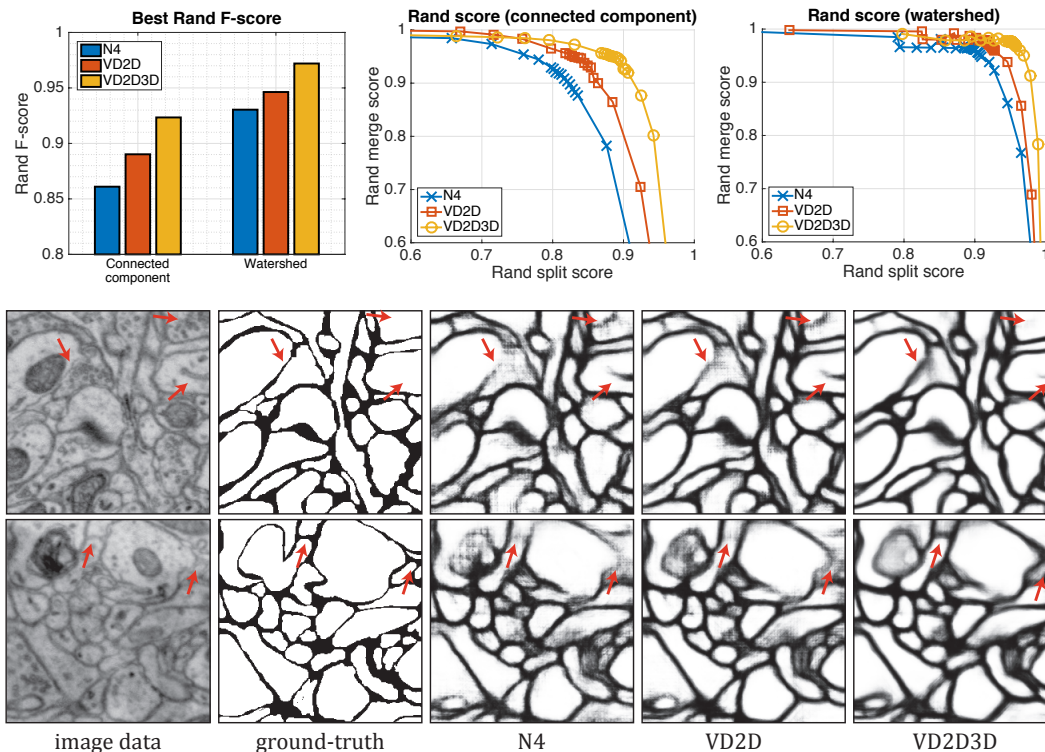

image data      ground-truth      N4      VD2D      VD2D3D

Figure 4: Quantitative (top) and qualitative (middle and bottom) evaluation of results.

which demonstrates the effectiveness of the proposed 2D-3D ConvNet combined with recursive approach. The middle and bottom rows in Figure 4 show some example locations in test set where both 2D models (N4 and VD2D) failed to correctly detect the boundary, or erroneously detected false boundaries, whereas VD2D3D correctly predicted on those ambiguous locations. Visual analysis on the boundary detection results of each model again demonstrates the superior performance of the proposed recursively trained 2D-3D ConvNet over 2D models.

# 7 Discussion

**Biologically-inspired recursive framework**      Our proposed recursive framework is greatly inspired by the work of Chen et al. [34]. In this work, they examined the close interplay between neurons in the primary and higher visual cortical areas (V1 and V4, respectively) of monkeys performing contour detection tasks. In this task, monkeys were trained to detect a global contour pattern that consists of multiple collinearly aligned bars in a cluttered background.

The main discovery of their work is as follows: initially, V4 neurons responded to the global contour pattern. After a short time delay ($\sim$40 ms), the activity of V1 neurons responding to each bar composing the global contour pattern was greatly enhanced, whereas those responding to the background was largely suppressed, despite the fact that those "foreground" and "background" V1 neurons have similar response properties. They referred to it as "push-pull response mode" of V1 neurons between foreground and background, which is attributable to the top-down influence from the higher level V4 neurons. This process is also referred to as "countercurrent disambiguating process" [34].

This experimental result readily suggests a mechanistic interpretation on the recursive training of deep ConvNets for neuronal boundary detection. We can roughly think of V1 responses as lower level feature maps in a deep ConvNet, and V4 responses as higher level feature maps or output activations. Once the overall 'contour' of neuonal boundaries is detected by the feedforward processing of VD2D, this preliminary boundary map can then be recursively fed to VD2D3D. This process

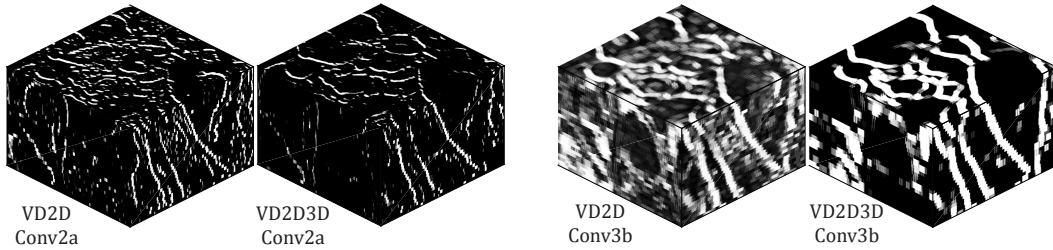

Figure 5: Visualization of the effect of recursive training. Left: an example feature map from the lower layer `Conv2a` in VD2D, and its corresponding feature map in VD2D3D. Right: an example feature map from the higher layer `Conv3b` in VD2D, and its corresponding feature map in VD2D3D. Note that recursive training greatly enhances the signal-to-noise ratio of boundary representations.

can be thought of as corresponding to the initial detection of global contour patterns by V4 and its top-down influence on V1.

During recursive training, VD2D3D will learn how to integrate the pixel-level contextual information in the recursive input with the low-level features, presumably in such a way that feature activations on the boundary location are enhanced, whereas activations unrelated to the neuronal boundary (intracellular space, mitochondria, etc.) are suppressed. Here the recursive input can also be viewed as the modulatory 'gate' through which only the signals relevant to the given task of neuronal boundary detection can pass. This is convincingly demonstrated by visualizing and comparing the feature maps of VD2D and VD2D3D.

In Figure 5, the noisy representations of oriented boundary segments in VD2D (first and third volumes) are greatly enhanced in VD2D3D (second and fourth volumes), with signals near boundary being preserved or amplified, and noises in the background being largely suppressed. This is exactly what we expected from the proposed interpretation of our recursive framework.

**Potential of ZNN**    We have shown that ZNN can serve as a viable alternative to the mainstream GPU-based deep learning frameworks, especially when processing 3D volume data with 3D ConvNets. ZNN's unique features including the large output patch-based training and the dense computation of feature maps can be further utilized for additional computations to better perform the given task. In theory, we can perform any kind of computation on the dense output prediction between each forward and backward passes. For instance, objective functions that consider topological constraints (e.g. MALIS [35]) or sampling of topologically relevant locations (e.g. LED weighting [15]) can be applied to the dense output patch, in addition to loss computation, before each backward pass.

Dense feature maps also enable the straightforward implementation of multi-level feature integration for fine-grained segmentation. Long et al. [30] resorted to upsampling of the higher level features with lower resolution in order to integrate them with the lower level features with higher resolution. Since ZNN maintains every feature map at the original resolution of input, it is straightforward enough to combine feature maps from any level, removing the need for upsampling.

### Acknowledgments

We thank Juan C. Tapia, Gloria Choi and Dan Stettler for initial help with tissue handling and Jeff Lichtman and Richard Schalek with help in setting up tape collection. Kisuk Lee was supported by a Samsung Scholarship. The recursive approach proposed in this paper was partially motivated by Matthew J. Greene's preliminary experiments. We are grateful for funding from the Mathers Foundation, Keating Fund for Innovation, Simons Center for the Social Brain, DARPA (HR0011-14-2-0004), and ARO (W911NF-12-1-0594).

## Footnotes

[1]Feature maps with the same resolution as the original input image. See Figure 5 for example. Note that the feature maps shown in Figure 5 keep the original resolution even after a couple of max-pooling layers.

[2]These layers are identical to `Conv1a`, `Conv1b`, and `Conv1c`.

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
