[Reviews · NeurIPS 2015]

Submitted by Assigned_Reviewer_1

The authors extend the winning segmentation algorithm from the ISBI 2012 segmentation challenge (Base2D in their paper) with two major extensions. Following advances in convnets for image recognition, the base architecture is made deeper, with smaller convolutions. The output from the 2D network is then used to initialize a 3D network, with further improvements in segmentation accuracy.

In terms of quality, the paper presents a clear advance in the state of the art on a challenging and important problem. The experiments use a number of metrics, both subjective and quantitative, to demonstrate their improvements. I would have liked to see experiments on a larger variety of datasets, and possibly larger datasets, since scaling is a major concern in connectomics applications. I would also hope future work more thoroughly compares against methods beyond the ISBI 2012 winner, including many of the papers they cite, and more recent advances such as this: http://arxiv.org/pdf/1506.04304.pdf

The paper is very clearly presented. There is not much in the way of dense derivations, since the architecture is a straightforward combination of existing parts. The figures are well-organized and thorough.

In terms of originality, some ideas are imported from other application areas (which is valuable in its own right, to see if those ideas can generalize beyond their original domain) while others derive inspiration from understanding of the biological visual system. None of the parts of this architecture are wildly original or complicated, which I see as a virtue, as it will be easy for practitioners in the field to adopt.

This paper is significant in that automating EM segmentation has the potential to revolutionize neuroscience. While this is an incremental advance in that direction, any improvements in efficiency are useful to the community and should be shared.
Summary: The authors present a novel convolutional neural network architecture for segmenting connectomics data. They present convincing evidence that their method leads to superior segmentation on this challenging and important problem.

Submitted by Assigned_Reviewer_2

The paper proposed to use a composition of a ConvNet with 2D filters and a ConvNet with 3D filters to improve the quality of neural boundary prediction. The authors motivate their architectural choice by the anisotropy (x-y vs z) of the sequential EM scans often used in Neuroscience. Their methods clearly outperforms the previous method (only using 2D filters) on this task which is important for their application.

The quality of the presentation could be easily improved and presentation could be streamlined. The method employs previously suggested techniques such as the recursive (composition) of nets and the application of 3D filters for 3D data sets.

The paper has unfortunately some problems. The paper refers often to the implementation ZNN (apparently theirs) that is mentioned to be crucial to get it efficient on the size of the network and the amount of data. However, there is no reference provided nor sufficient details are provided to reproduce it. I suggest to actually provide concise details on the essential parts of the implementation and also on whether the software will be released in some way. Otherwise the paper is full of general design choices used to develop ZNN that are not precise enough. The captions to the figures should be more descriptive, as important information is missing to understand them.

Details: l80: no reference for ZNN is provided. The reader has no idea what ZNN is. Is it an algorithm or a particular implementation? l177: the description of the Rand index is not sufficiently clear. There is a reference, but the paper should be self-consistent. Please provide the necessary formula. Section 3: This section does not really help to understand the paper. ZNN is not described in enough detail to reproduce the results. I suggest to give more precise details, or provide a reference to the resource where details can be found. Figure 3: color code? l393: In Fig 1 and the rest of the paper you have not used net1 and net2, please be consistent l396: As a comment: recursive seems not to be the right term for because it is not the same network that is recursively applied. It is the composition if vd2d an vd2d3d, although they share the same first layers. Is the vd2d part actually the same or just copied after the initial training and then refined independently?

Summary: The paper is relevant, but rather applied. The presentation needs improvements.

Submitted by Assigned_Reviewer_3

In this paper the authors discuss a new deep convolutional network employed for for segmentation of neurons in EM image stacks. The novelty fo teh paper is the arrangement of filters in quite deep networks and the inclusion of 3D filters. Overall the paper is of good quality and is well-written/clear. The authors appear to have improved on previous methods and the application area is quite important for neuroscience. Overall I think it is worth presenting even if there are not necessarily any new fundamental insights about the method (other than the detailed architecture).
Summary: A good paper even if the novelty appears to be a bit in the details.

Submitted by Assigned_Reviewer_4

This paper proposes a 2D-3D convolutional network to detect neuronal boundary from 3D electron microscopic (EM) brain image. Different from the traditional convolutional network, the proposed method uses both 3D and 2D filters to encode the information in EM image sequence. In the training stage, a recursive schema is applied to effectively learn 3D features from 2D initialization. Further, ZNN is applied for more efficient computation. The proposed method is tested on cortex images. Compared with base2D and VD2D models, the proposed 2D-3D model achieves lower error rate.

The paper is in general well written and easy to understand. The proposed method shows novelty in neuronal boundary detection and has potential to be applied to other sequential data such as video frames.

However, the experiment part is not strong enough. Although 2D-3D model demonstrates more accurate boundary detection compared with 2D models, further experiments should be carried out, such as the comparison experiments against the state-of-the-art in boundary detection.

Summary: The paper is interesting and generally well written. More comparison to the state-of-the-art in neuronal boundary detection is needed to demonstrated the power of the proposed method in such an application.

Submitted by Assigned_Reviewer_5

The subject matter of the paper is very far out of my area of expertise. I found the paper clear and the motivations for the architecture convincing and intuitive, and the results support the hypothesis, so it seems like a good contribution. I am incapable of giving a more in-depth analysis than this, I'm afraid, and encourage the committee and authors to rely more on the other reviews.
Summary: This paper provides a novel architecture for the detection of neuronal boundaries in 3d scans. The main contribution is the interpolation of boundary classifications from 3d convnets and 2d convenets, which, combined, produce improved results.

Author Feedback
Author rebuttal: We are grateful for the positive and informative reviews. R2 wrote that "The novelty for the paper is the arrangement of filters in quite deep networks and the inclusion of 3D filters." R3 wrote that "the paper presents a clear advance in the state of the art on a challenging and important problem." R3 also wrote that "This paper is significant in that automating EM segmentation has the potential to revolutionize neuroscience."

Since our approach combines several elements, the novelty is somewhat complicated to express, and worth repeating here. The hybrid use of 2D and 3D filters in ConvNet is novel. The use of 3D filters is novel for highly anisotropic images produced by serial section EM. This is the first application of the recursive ConvNet approach to boundary detection (though see related work of Huang & Jain 2014 on a recursive non-ConvNet approach, and see discussion of the definition of "recursive" later on).

The major reservations were expressed by R1 and R4. R4 expressed the reservation that ZNN was not described in sufficient detail to replicate the results, and asked whether the software will be released. ZNN was already publicly released months before the submission, but the link was omitted to preserve anonymity. R4 and other experts should be able to replicate our results with the publicly available source code, based on the information given in the paper. We apologize for the confusion caused by the anonymity requirement.

R1 questioned whether the baseline method (IDSIA's ConvNet) remains representative of the state-of-the-art. This is a reasonable question given that three years have passed since IDSIA won the ISBI'12 challenge. It turns out that IDSIA can be compared to more recent methods that were submitted to the challenge site after 2012. Some submissions are ranked higher than IDSIA on the official leaderboard (though many of them applied postprocessing to IDSIA's boundary map). However, the challenge organizers have recently discovered that their scoring system was insufficiently robust to variations in the thickness of neuronal boundaries. Once this weakness is corrected, it turns out that there are no significant gains since 2012 (personal communication with ISBI'12 challenge organizers). Therefore, it appears that IDSIA's ConvNet remains state-of-the-art in neuronal boundary detection, for the specific case of highly anisotropic serial section EM images. This is why we employed IDSIA's N4 architecture (we will replace Base2D with N4 in the revised manuscript) as a baseline for performance comparison.

Having addressed the major reservations, we move on to other questions posed by the reviewers. R3 was interested in comparisons with supervoxel agglomeration methods, specifically mentioning Maitin-Shepard et al. 2015, http://arxiv.org/pdf/1506.04304.pdf. In our opinion, boundary detection methods do not compete with supervoxel agglomeration methods, because they are complementary. Pipelines for 3D reconstruction of neurons generally employ (1) an initial stage of boundary detection to generate supervoxels, followed by (2) some method of agglomerating supervoxels to form neurons. Therefore it is likely that improving a boundary map will also improve the results of supervoxel agglomeration. For this reason, we feel that comparison with supervoxel agglomeration methods is outside the scope of the present paper. Furthermore, features learned in the boundary detection stage are often used as input to supervoxel agglomeration. Our method learns 3D features, which would not be available from a purely 2D boundary detection algorithm.

Lastly, we'd like to answer the detailed questions raised by R4. One of them asks whether ZNN is "an algorithm or a particular implementation." From the machine learning viewpoint, ZNN is an implementation of 3D ConvNets. From the parallel computing viewpoint, ZNN contains a new algorithm that splits ConvNet training into tasks and schedules them across a set of workers so as to maximize utilization. To achieve high performance and scalability, a dedicated memory management algorithm is developed together with a CPU-specific scheduler. Given its complexity, the parallel algorithm is outside the scope of the present paper and will be described in detail in a separate publication.

Another question asks whether it is accurate to use the term "recursive". The 2D part of VD2D3D is initialized with learned weights of the corresponding part of VD2D, and then we fine-tuned the whole VD2D3D. Therefore the 2D part of VD2D3D does not have the same weights as VD2D, and is not recursive in the precise sense. However, the 2D part of VD2D3D does have the same architecture as VD2D, so we feel that the term "recursive *training*" is fairly accurate if not perfect. We will change the title to "Recursive Training of 2D-3D Convolutional Network for Neuronal Boundary Detection", and will modify related content accordingly in the revised manuscript.